# Towards Tree Green Crown Volume: A Methodological Approach Using Terrestrial Laser Scanning

**Zihui Zhu, Christoph Kleinn and Nils Nölke \***

Forest Inventory and Remote Sensing, Faculty of Forest Sciences and Forest Ecology, University of Göttingen, Büsgenweg 5, 37077 Göttingen, Germany; zzhu@gwdg.de (Z.Z.); ckleinn@gwdg.de (C.K.)

\* Correspondence: nnoelke@gwdg.de

**Abstract:** Crown volume is a tree attribute relevant in a number of contexts, including photosynthesis and matter production, storm resistance, shadowing of lower layers, habitat for various taxa. While commonly the total crown volume is being determined, for example by wrapping a convex hull around the crown, we present here a methodological approach towards assessing the tree green crown volume (*TGCVol*), the crown volume with a high density of foliage, which we derive by terrestrial laser scanning in a case study of solitary urban trees. Using the RGB information, we removed the hits on stem and branches within the tree crown and used the remaining leaf hits to determine *TGCVol* from $k$-means clustering and convex hulls for the resulting green 3D clusters. We derived a tree green crown volume index (*TGCVI*) relating the green crown volume to the total crown volume. This *TGCVI* is a measure of how much a crown is "filled with green" and scale-dependent (a function of specifications of the $k$-means clustering). Our study is a step towards a standardized assessment of tree green crown volume. We do also address a number of remaining methodological challenges.

**Keywords:** urban trees; green crown volume; point cloud; $k$-means

## 1. Introduction

Crowns are important tree components, as they produce oxygen, offer habitats for many taxa, filter out dust and other pollutants, generate shadow and do largely determine the scenic beauty of trees and forests. However, tree crowns and the variables characterizing them are difficult to define and measure; that holds for any crown variable, be it wood volume, leaf area, crown volume, crown projection area, etc. Additionally, all these crown variables are among those forest mensurational variables for which "true values" are virtually impossible to determine (at the standing tree and non-destructively). The crown, therefore, while being the most important part of the tree (engine), poses fundamental challenges for forest and tree mensuration. Among crown variables, the crown projection area is a straightforward measure that can relatively easily be defined and determined, thus there is a long tradition using it to describe tree crowns: it is used as a competition metric [1,2], to establish models to predict, for example, biodiversity [3] or when dealing with trees outside the forest (TOF); in addition, fragmentation metrics of such crown polygons are used to characterize the pattern of tree spatial distribution [4]. In forest definitions, canopy cover derived from individual trees' crown cover is frequently used as a core criterion [5].

One may ask the question whether the crown projection area alone is always a sufficiently exhaustive measure to characterize tree crowns, as, for example, for one and the same crown projection area, tree crowns may have very different 3D crown shapes, volumes and densities. Assessment and modeling of crown variables in the third dimension pose additional challenges: in the first and very basic place, this refers to challenges of definitions [6] and measurement protocols.

An interesting approach regarding both definition and measurement is what the artists Christo and Jeanne-Claude did when they wrapped tree crowns with large sheets of woven polyester fabric in the years 1997–1998 in Basel, Switzerland [7]. What these envelopes wrap is the total crown volume of the entire crown including all permanent woody crown components, leaves and empty spaces. By using such envelopes, the artists used an implicit definition of total crown volume.

Although difficult to define, there are a number of applications in which 3D crown variables may be of interest. When modeling, for example, the tree crown as a habitat for taxa like insects, birds and bats, it is likely that the habitat quality of a tree crown will also be a function of the volume "filled by leaves" and its distribution within the crown. So far, to avoid the additional challenges, most of the studies using instead the leaf area index (LAI) [8] for modeling as a common measure to quantify leaf density, but it is again a 2D "summary" of foliage with the advantage that there is a straightforward definition (leaf area per unit horizontal surface area). LAI is commonly not done by measuring leaf areas, but through optical devices or cameras devices that determine measure variables like crown density or crown transmissibility which is then used as a proxy and translated into values of LAI.

With terrestrial laser scanning (TLS), there is a technology that has the potential to receive the 3D structure of a tree crown. Several studies made use of these dense 3D point clouds to retrieve foliage attributes. The most common one is the leaf area density (LAD) which is as the LAI straightforward defined as the one-sided area of leaves per unit volume. To estimate LAD, voxelization approaches are used with the weakness that the size of a voxel is quite sensitive [9].

In this study, we introduce TLS-based *k*-means clustering as a proxy for a new 3D crown variable named tree green crown volume (*TGCVol*). *TGCVol* is certainly one of those crown variables that are difficult to define and difficult to assess; and research needs to resort to proxies to make empirical studies feasible and to the best knowledge of the authors, *TGCVol* has so far not been introduced nor assessment approaches presented. The goal of this study is to develop and evaluate a TLS-based approach to assess the *TGCVol*, to describe its scale dependency and discuss challenges regarding definitions, measurements and analyses. Our technical objectives are: (a) to identify the crown leaf hits from merged multiple scans of solitary trees during the vegetation period; (b) to derive *TGCVol* from extracted leaf hits from a *k*-means clustering approach; (c) to illustrate the scale dependency of *TGCVol* by varying the *k* in *k*-means clustering.

## 2. Materials and Methods

### 2.1. Study Area

Data were collected within the city of Göttingen, Germany. Twenty-six sample trees were selected (Figure 1) such that they cover a wide range of tree species, crown shapes and were well visible from all sides for a straightforward observation by terrestrial laser scanning. The sample trees were located on roadsides, in botanical gardens and in parks. Photographs of a subset of sample trees are given in Figure 2 for illustration.

### 2.2. Workflow Outline

The workflow for determining *TGCVol* is shown in Figure 3. It has one main processing line of point cloud analyses. In a final step (see Section 2.7), an easy-to-use index was derived to predict *TGCVol*. The processing and analysis workflow was implemented on a high-performance computer.

### 2.3. Definition

While the basic idea behind *TGCVol* is easily described as "the sum of spaces in the crown filled with leaves", it turns out to be difficult to come up with an unambiguous definition that may form the basis for a likewise unambiguous measurement protocol. This is not uncommon in forest mensuration and forest inventory where there are many variables that pose alike issues. In particular, other crown variables bring the same challenge with respect to measurement protocols, including the variables

leaf area density (LAD) and leaf area index (LAI). For both variables, a theoretical definition can be formulated (LAD = sum of one-sided green leaf area per unit volumes; LAI = sum of one-side green leaf area per horizontal unit area) but it is hardly possible to measure them directly: their observation is, therefore, based on the assessment of meaningful proxies. For LAD, when proxied by TLS scans, it is commonly so that the 3D points are first converted into a 3D voxel array of an arbitrary size, an approach highly sensitive to voxel size [9,10]. With the voxel information provided, different methods exist to assess LAD, e.g., [9,11]. Following the basic idea, *TGCVol* is proxied in this study by *k*-means convex hull clustering, applied, for example, also in the field of object recognition [12]: we use the green TLS hits identified by the RGB information to recognize "leaf clusters". *TGCVol* is then defined as the sum of all the volumes of envelopes of "leaf clusters".

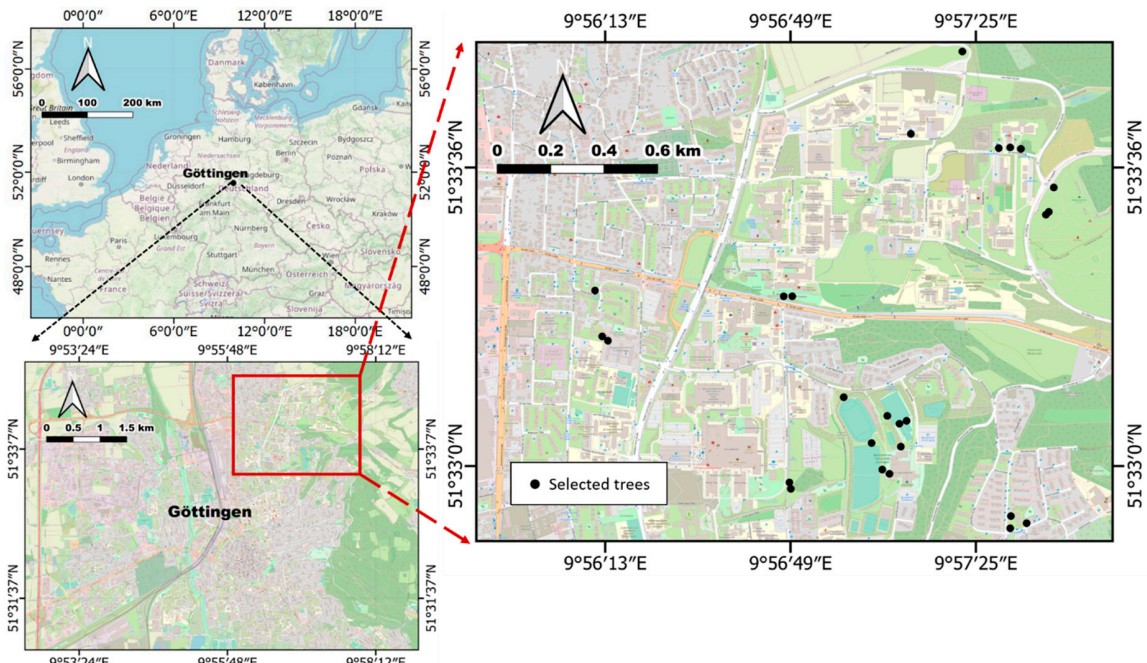

**Figure 1.** Study area and the locations of the selected trees (map source: © OpenStreetMap contributors).

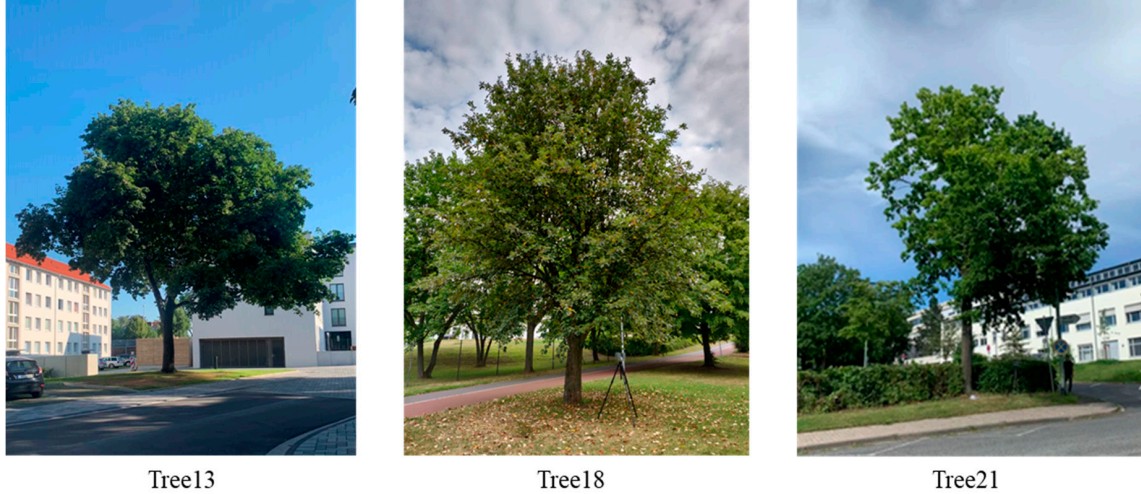

**Figure 2.** Illustration of the variation of crown sizes, shapes and densities among the sample trees.

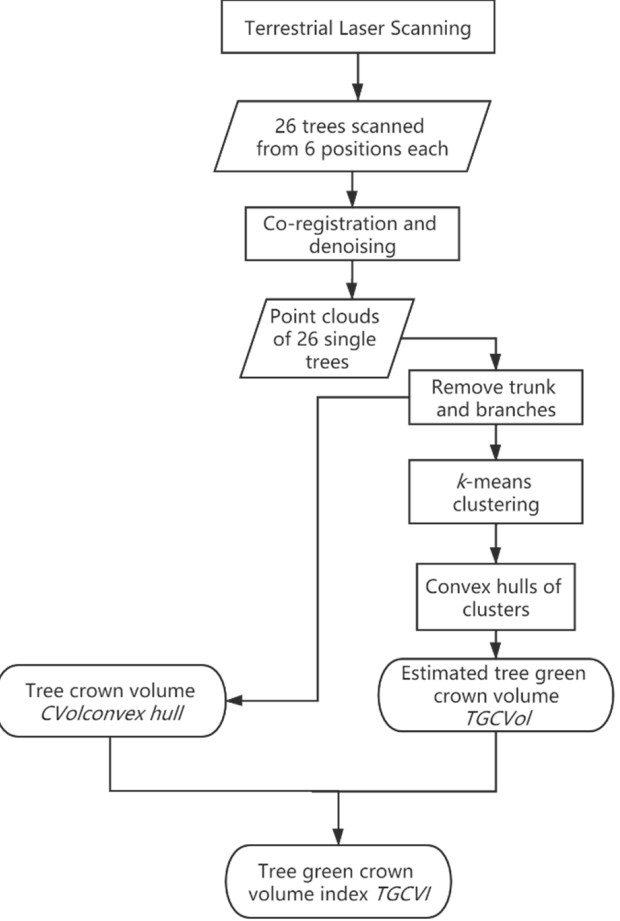

**Figure 3.** Schematic workflow.

*2.4. Field Measurements*

Table 1 lists the terms and their definitions used in the field inventory.

**Table 1.** Terms used in this study.

| Terms | Elements of the Definition | Notation Here |
|---|---|---|
| Crown base height | The vertical distance from ground to the lowest leaf layer of the tree crown meaning the first foliage advancing upwards from the ground. | *CBH* |
| Crown length | The vertical distance between crown base and crown (tree) top. | *CL* |
| Crown width | The width of horizontal tree crown projections in one direction. Here, we used two fixed directions and averaged the measurements. | *CW* |
| Tree height | The vertical distance from the ground level to the level of the crown (tree) top. | *H* |
| Diameter at breast height | The diameter of the stem at the height of 1.3m above ground measured perpendicularly to the stem axis. | *DBH* |

Per tree, field measurements and scanning were done on the same day. For each sample tree, the following variables were measured: *DBH* by diameter tape; total tree height *H* and crown base height *CBH* by a Vertex® IV hypsometer (Haglöf, Sweden). Crown width *CW* was determined by tape in two fixed directions (South-North and East-West), following the definitions as of Table 1.

Table 2 lists the descriptive statistics of the major dendrometric variables of the sample trees. Crown width *CW* is the average of two measurements. The set of sample trees embraced a total

of 10 tree species with a *DBH* and an averaged *CW* ranging from about 20 cm to 90 cm and 6 m to 18 m, respectively.

**Table 2.** The sample trees and their major crown-relevant characteristics.

| Tree Id | Species | Location | DBH (cm) | H (m) | CBH (m) | Averaged CW (m) |
|---|---|---|---|---|---|---|
| Tree1 | *Acer platanoides* | Botanical Garden | 35.8 | 12.5 | 1.0 | 7.4 |
| Tree2 | *Acer platanoides* | Botanical Garden | 34.0 | 10.5 | 1.5 | 6.2 |
| Tree3 | *Acer pseudoplatanus* | Roadside | 58.0 | 17.1 | 0.7 | 11.6 |
| Tree4 | *Acer pseudoplatanus* | Roadside | 61.2 | 14.8 | 0.9 | 11.85 |
| Tree5 | *Acer pseudoplatanus* | Roadside | 46.6 | 12.8 | 0.8 | 12.45 |
| Tree6 | *Acer pseudoplatanus* | Roadside | 35.6 | 10.9 | 1.0 | 12.0 |
| Tree7 | *Acer pseudoplatanus* | Roadside | 43.8 | 17.4 | 1.6 | 13.7 |
| Tree8 | *Acer pseudoplatanus* | Roadside | 88.2 | 22 | 1.0 | 13.9 |
| Tree9 | *Acer pseudoplatanus* | Roadside | 49.5 | 10.7 | 1.0 | 13.65 |
| Tree10 | *Acer pseudoplatanus* | Roadside | 43.0 | 10.5 | 2.0 | 11.25 |
| Tree11 | *Acer pseudoplatanus* | Roadside | 16.9 | 14 | 1.8 | 6.5 |
| Tree12 | *Acer pseudoplatanus* | Roadside | 20.4 | 13.6 | 2.4 | 7.75 |
| Tree13 | *Acer pseudoplatanus* | Roadside | 55.8 | 15.3 | 2.6 | 14.25 |
| Tree14 | *Acer pseudoplatanus* | Roadside | 36.2 | 11.6 | 2.1 | 10.85 |
| Tree15 | *Aesculus hippocastanum* | Roadside | 72.8 | 18.2 | 2.2 | 14.3 |
| Tree16 | *Aesculus × carnea* | Botanical Garden | 20.0 | 6.3 | 1.0 | 6.35 |
| Tree17 | *Carpinus betulus* | Roadside | 53.6 | 14.6 | 0.5 | 11.5 |
| Tree18 | *Fagus sylvatica* | Roadside | 42.7 | 12.5 | 1.5 | 8.3 |
| Tree19 | *Liriodendron tulipifera* | Roadside | 39.5 | 16.8 | 0.0 | 10.35 |
| Tree20 | *Prunus avium* | Botanical Garden | 42.6 | 8.7 | 1.5 | 8.0 |
| Tree21 | *Quercus robur* | Roadside | 43.5 | 12.4 | 2.4 | 10.6 |
| Tree22 | *Quercus robur* | Roadside | 64.1 | 15.4 | 1.4 | 15.8 |
| Tree23 | *Quercus robur* | Roadside | 59.1 | 12.8 | 1.0 | 17.35 |
| Tree24 | *Quercus robur* | Roadside | 29.4 | 12.4 | 3.0 | 12.2 |
| Tree25 | *Tilia cordata* | Roadside | 43.5 | 15.0 | 2.0 | 11.15 |
| Tree26 | *Tilia cordata* | Roadside | 59.9 | 16.1 | 1.4 | 11.6 |

*2.5. Terrestrial Laser Scanning*

A Trimble® TX5 laser scanner (Trimble Navigation, Ltd., USA) was used, with a scan resolution of 177Mpts per full scan, resulting in a scan duration of 3'35" and a point spacing of approximately 3mm at a distance of 10m. Additionally, RGB information was recorded per hit using the color image capture option which extended the scanning duration up to an average of 6'35". Scans were acquired during July and August 2019 in the midst of the vegetation period. The moderate and strong wind (wind speed higher than 14 km/h) was avoided as far as possible during the scanning process.

We used a multi-single-scan method where each sample tree was scanned from 6 positions (Figure 4) to guarantee a detailed 3D representation of the outer and inner parts of the crowns. Three of the positions were chosen in distances from the tree, that allow obtaining the returns from the crown surface to the extent of possible, ranging between 5m and 20m. The remaining 3 scanning positions were located close to the stem, ranging between 0.5m and 1m, beneath the tree crown to guarantee that most hits came from the inner part of the crown. The 6 single scans per tree were merged with a maximum registration error of less than 6mm by automatically cloud-to-cloud matching using Autodesk® ReCap V5.0.4.17, this procedure repeated until all the corresponding scans were merged. The laser returns of the single trees were manually extracted in CloudCompare v2.10.2. We used the CloudCompare's "statistic outlier remover" SOR (PCL, 2011) [13] to remove the noise and exported each single tree as a ".las" file for further processing in R (R Core Team, 2019).

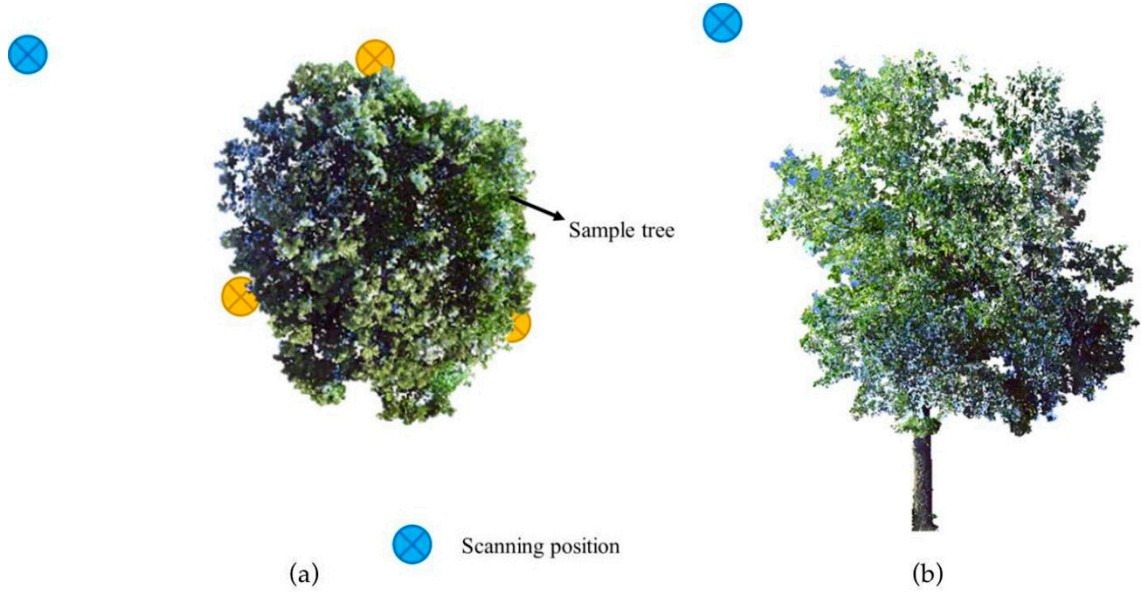

**Figure 4.** (**a**) Location of the laser scanner (view from above) relative to the stem position, blue circles represent the scanning positions far from the tree, yellow circlets represent the scanning positions close to the stem; and (**b**) extracted point cloud (view from the front). Sample Tree 21.

### 2.6. Determining of Tree Green Crown Volume

Our approach to observe *TGCVol* included a number of processing steps (Figure 3). To remove the woody elements from the point cloud (stem and branches), we used the CANUPO classification algorithm [14] implemented in CloudCompare. The CANUPO classification is a multiscale dimensionality analysis to characterize features according to their geometry and RGB information [14]: the classifier is trained by small samples and then applied to a point cloud to separate it into two categories, here: "woody parts" and "rest". Training data were sampled to cover various cases that can be encountered for each category, for example, over- or underexposure. In a visual validation, the outcome of this classification turned out to be incomplete: various woody hits still remained. Manual point cloud editing needed therefore to be applied and hits on woody elements were visually identified from their RGB information and then manually removed from the point cloud (Figure 5).

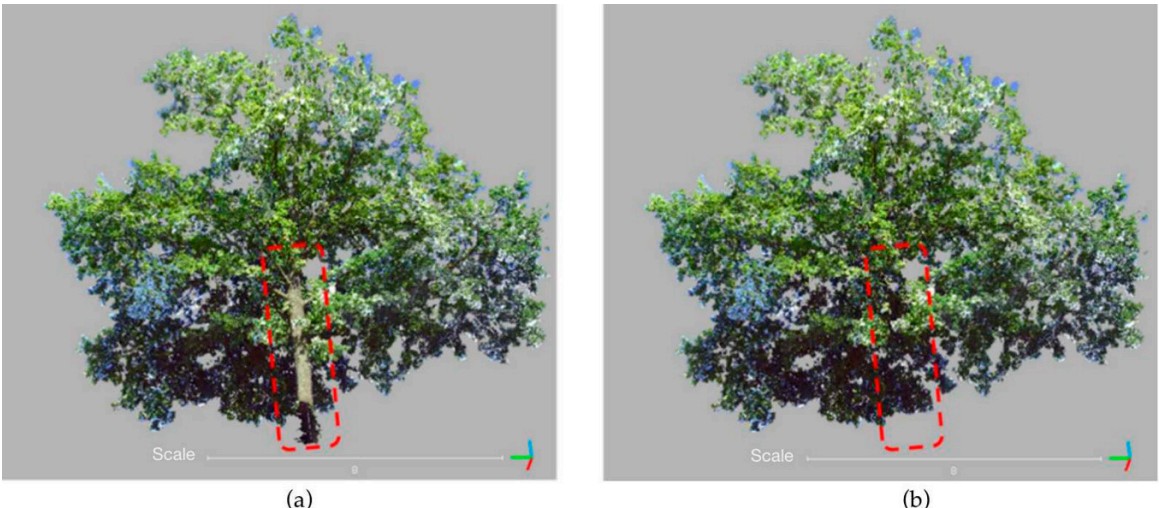

**Figure 5.** Point clouds of sample Tree 21 (view from below): (**a**) original point cloud; (**b**) remaining points after removing trunk and branches by automatic plus manual removal.

The *k*-means clustering approach [15] was then applied to cluster groups of nearby leaf hits using R package *Morpho* [16]. *k*-means clustering is a technique that comes from signal processing and received a variety of applications: market segmentation, document clustering, and image segmentation, etc. It involves iteratively exploring the centroid of a cluster of points and then regrouping the neighboring points into this cluster where *k* is the initial parameter that represents the total number of clusters to be produced. The distance between leaf hits in each cluster is indirectly defined *k*-means clustering approach: a smaller value of *k* produces lesser but larger clusters allowing a longer distance between leaf hits to be included into the same cluster, while a large value of *k* generate many smaller clusters and demand a shorter maximum distance for leaf hits to be clustered. We started always with $k = 1$. The convex hull of this one cluster is wrapping the total crown volume $CVol_{convex\ hull}$ (see Figure 6a). With each further iteration, we increased *k* so that more and smaller clusters were generated separating more and more the green and empty spaces within the crown (Figure 6). Around each cluster of hits, a convex hull was wrapped by using R package *rLiDAR* [17]. The *TGCVol* is calculated by summing up all the volumes of all these convex hulls. We increased *k* stepwise up to the value of $k = 1400$. For each value of *k*, the sum of the wrapped clusters constitutes the *TGCVol* at this particular spatial resolution. We then plotted the resulting *TGCVol* over *k* for interpretation of the scale dependency. A suitable or even optimal value of *k* will depend on the specific subject matter objective of a study and is not a focus of this research.

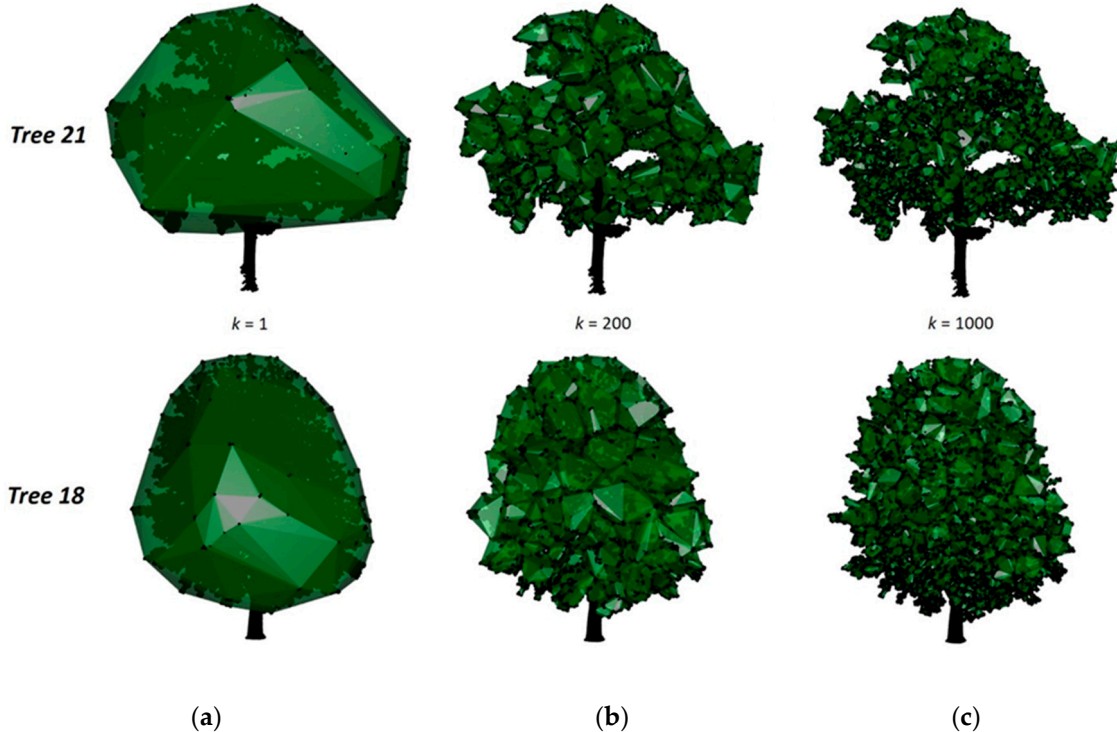

(**a**)　　　　　　　　　　　(**b**)　　　　　　　　　　　(**c**)

**Figure 6.** Illustration of the *k*-means clustering approach to generate hulls of green volume within the crown (sample Tree 21 and Tree 18): convex hulls were wrapped around: (**a**) $k = 1$ cluster (wrapping total crown volume); (**b**) $k = 200$ clusters; and (**c**) $k = 1000$. "Leaf clusters" can be seen more evenly distributed in the crown of Tree 18 than of Tree 21.

*2.7. The Tree Green Crown Volume Index*

To make the values comparable between trees of different sizes, and as a simple measure of the degree to which the crown contains green volume, we developed the tree green crown volume index

*(TGCVI)* which is the percentage of tree green crown volume *(TGCVol)* within the total crown volume $CVol_{convex\ hull}$ ($k = 1$):

$$TGCVI = \frac{TGCVol}{CVol_{convex\ hull}}, \tag{1}$$

*TGCVI* tends towards a value of 1 when leaves occur uniformly all over the crown at a minimum density.

## 3. Results

When refining the separation of green and empty spaces by increasing the number of clusters *k*, the green crown volume decreases. This describes the scale dependency of determining *TGCVol*. For our sample trees, *TGCVol* decreased rapidly and then leveled out for values of *k* beyond 200–300. Here, the trends for Tree 18 was slower compared to Tree 21 (see also Figure 7).

Table 3 records the crown volume and tree green crown volume index *(TGCVI)* for the sample trees. The tree green crown volume index decreased with the increasing number of convex hulls *k*. The mean *TGCVI* was 0.58, 0.35 and 0.27 with standard deviations of 0.083, 0.082 and 0.075, respectively in the cases when *k* = 100, 500 and 1000. The small size of the study does not allow further analyses of differences between species, but we see in our results that for *Quercus robur* (sample trees 21 to 24) the values of *TGCVI* were relatively close together (0.54 to 0.45, 0.28 to 0.23 and 0.20 to 0.18), and consistently lower than the other species with similar dimensions.

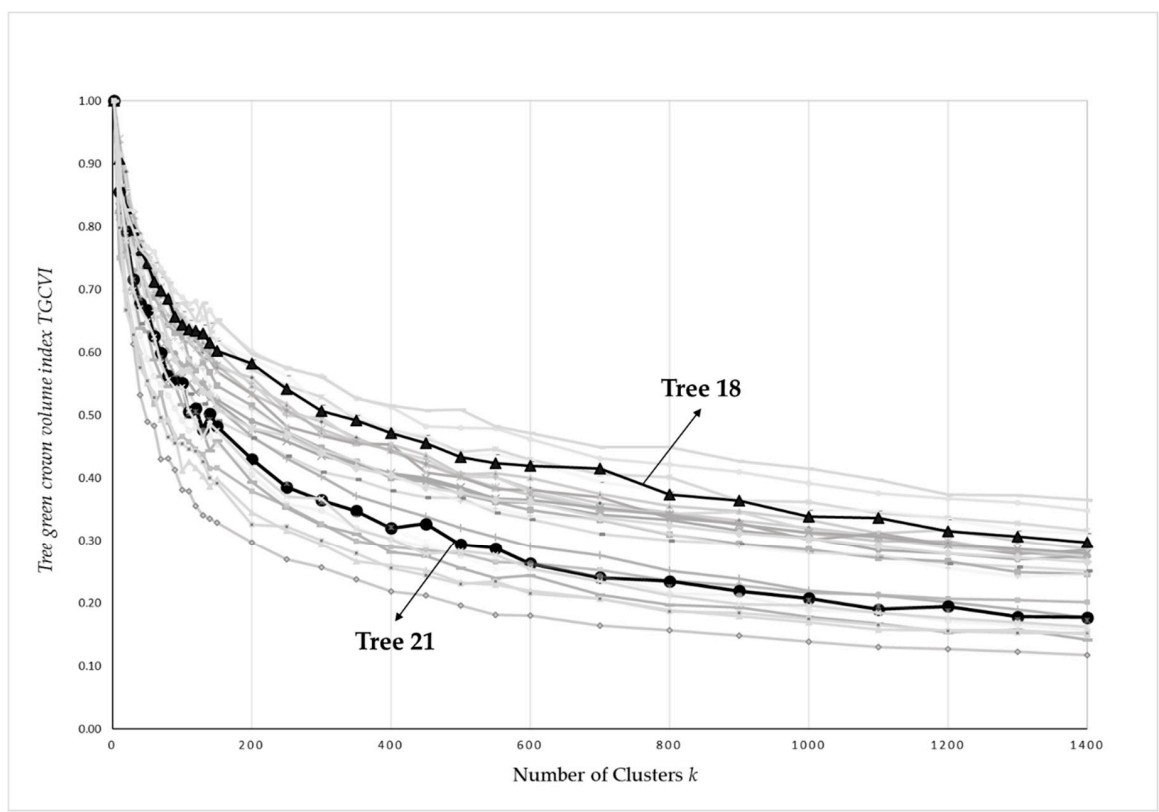

**Figure 7.** Tree green crown volume index *(TGCVI)* over number of clusters *k*. As to be expected: the more clusters are formed, the finer is the separation of green and empty spaces within the crown which leads to decreasing values of *TGCVI*. The *y*-axis is the normalized *TGCVI*. Tree 21 and Tree 18 are highlighted by the bold line with markers.

**Table 3.** The sample trees and their tree green crown volume index.

| Tree Id | Species | $CVol\;_{convex\;hull}$ **(m³)** | TGCVI When $k =$ | | |
|---|---|---|---|---|---|
| | | | **100** | **500** | **1000** |
| Tree1 | *Acer platanoides* | 289 | 0.63 | 0.41 | 0.32 |
| Tree2 | *Acer platanoides* | 196 | 0.63 | 0.37 | 0.29 |
| Tree3 | *Acer pseudoplatanus* | 1217 | 0.66 | 0.41 | 0.33 |
| Tree4 | *Acer pseudoplatanus* | 990 | 0.62 | 0.40 | 0.30 |
| Tree5 | *Acer pseudoplatanus* | 1026 | 0.66 | 0.44 | 0.36 |
| Tree6 | *Acer pseudoplatanus* | 700 | 0.58 | 0.38 | 0.30 |
| Tree7 | *Acer pseudoplatanus* | 1576 | 0.64 | 0.40 | 0.32 |
| Tree8 | *Acer pseudoplatanus* | 1736 | 0.66 | 0.43 | 0.35 |
| Tree9 | *Acer pseudoplatanus* | 812 | 0.52 | 0.26 | 0.18 |
| Tree10 | *Acer pseudoplatanus* | 473 | 0.38 | 0.20 | 0.14 |
| Tree11 | *Acer pseudoplatanus* | 163 | 0.47 | 0.28 | 0.22 |
| Tree12 | *Acer pseudoplatanus* | 270 | 0.41 | 0.23 | 0.17 |
| Tree13 | *Acer pseudoplatanus* | 1234 | 0.56 | 0.39 | 0.31 |
| Tree14 | *Acer pseudoplatanus* | 681 | 0.65 | 0.40 | 0.31 |
| Tree15 | *Aesculus hippocastanum* | 1814 | 0.69 | 0.48 | 0.39 |
| Tree16 | *Aesculus × carnea* | 111 | 0.58 | 0.32 | 0.22 |
| Tree17 | *Carpinus betulus* | 817 | 0.55 | 0.37 | 0.28 |
| Tree18 | *Fagus sylvatica* | 380 | 0.64 | 0.43 | 0.34 |
| Tree19 | *Liriodendron tulipifera* | 1049 | 0.57 | 0.38 | 0.30 |
| Tree20 | *Prunus avium* | 347 | 0.55 | 0.29 | 0.21 |
| Tree21 | *Quercus robur* | 620 | 0.45 | 0.23 | 0.18 |
| Tree22 | *Quercus robur* | 2092 | 0.54 | 0.28 | 0.20 |
| Tree23 | *Quercus robur* | 1729 | 0.54 | 0.28 | 0.20 |
| Tree24 | *Quercus robur* | 730 | 0.49 | 0.28 | 0.19 |
| Tree25 | *Tilia cordata* | 904 | 0.61 | 0.36 | 0.28 |
| Tree26 | *Tilia cordata* | 1003 | 0.68 | 0.51 | 0.42 |

## 4. Discussion

Our study is to be seen as a pilot study to further develop measurement and analysis approaches towards a better description of tree green crown volume. The limited number of 26 sample trees of 10 species in one single environment (urban trees) and without considering many different crown shapes does not allow further inferences about factors that determine amount and pattern of *TGCVol*; rather, our study was to introduce the concept of *TGCVol* to better describe the foliage distribution in tree crowns and present a first case study for its assessment, identifying remaining methodological challenges. We chose to do this first study on solitary trees in order to reduce the number of confounding features. We are aware, of course, that applications to trees in closed-canopy forests will pose a series of additional challenges. We have started with a simple condition but we will continue seeking better alternatives to enhance the applicability of our approach in our future work.

The scanning positions are an important element for the measurements as they need to be determined such that the whole crown can duly be scanned. Multiple scanning positions were necessary to avoid scan shadows and to capture the foliage information; this was challenging under the given conditions in an urban environment, mainly because of buildings and car traffic. Within a closed stand, though, the challenges will even be greater than with the solitary trees in our study because the visibility of the scanner may be blocked by neighboring trees. Further difficulties exist in the extraction of individual crowns. The scanning positions close to the tree will produce denser point clouds and allow identifying more details, but they come along with larger differences in hit density for closer and more distant crown parts. While this challenge can be overcome for the lower crown parts by multiple scan positions around the crown, it will remain for the upper crown parts (as partly seen in Figure 6c): there, the overall hit density is always lower which may affect the clustering analysis and lead to an overestimation of tree green crown volume.

When separating leaf and branch hits within the crown, the CANUPO classification that we applied here made it to segment most of the branches but there were still smaller branches that had to be removed manually. This was time-consuming and does certainly leave room for optimization. The branches that we needed to remove manually from the crown point cloud were all characterized by under- or overexposure of their RGB values, which is likely the reason that the automatic segmentation approach failed. This is a well-recognized and common problem in terrestrial laser scanning and not solved yet [18]. The RGB data is core to our approach to identifying leaf hits in the crown: the distinction between the leaf and branch hits will hardly be possible without these RGB data. Besides CANUPO which is easy-to-use and already implemented in CloudCompare, there are also alternatives for wood-leaf classification such as machine learning techniques [19], and we might improve the classification efficiency by seeking such an automated separation procedure [20].

*TGCVol* is usually not evenly distributed within the crown but comes in a clustered pattern. Then, it is obvious that determining *TGCVol* is scale-dependent. This is clearly illustrated in Figure 7 when plotting *TGCVol* over the number of clusters: the finer the separation between green and empty spaces, the smaller the overall *TGCVol*. The scale is here derived from the number of clusters, which implicitly defines how fine the clustering is. As with other scale-dependent features (like "forest structure", for example [21] or "forest edge length", for example [22]), the definition of a scale must be part of the definition of the features to make it unambiguous. The definition of a suitable scale for a particular purpose needs to come from subject matter considerations and cannot be generalized. Of course, the computational capacity must also be considered: the more detailed the clustering, the more computer power and processing time is required.

Figure 7 illustrates not only the scale dependency but also the differences between the sample trees. The decreasing trends of *TGCVol* point to differences in 3D distributions of green volume within the tree crowns. The lower curves (e.g., Tree 21) indicate crowns with a sparser distribution of smaller leaf clusters while the upper curves (e.g., Tree 18) point to more evenly arranged green volume or and large green clusters. A more comprehensive comparison also of more extreme crown architectures (very dense and very sparse crowns, pillar-shaped crowns and umbrella-shaped crowns, for example) will likely reveal more possibilities for the interpretation of the curves as of Figure 7.

## 5. Conclusions

This study presented a TLS-based approach to assess tree green crown volume (*TGCVol)*. We see many useful applications of *TGCVol* in particular in the context of trees outside the forest, for example in the modeling of urban trees for habitat suitability and heat mitigation, as a lower green crown volume will probably result in heat mitigation and influence the nesting site selection of taxa; but we also acknowledge (and addressed it in this paper) that there are numerous methodological challenges that wait to be resolved.

**Author Contributions:** Conceptualization, C.K. and N.N.; methodology, N.N. and Z.Z.; software, N.N. and Z.Z.; formal analysis, Z.Z.; data curation, Z.Z.; writing—original draft preparation, Z.Z.; writing—review and editing, C.K., N.N. and Z.Z. All authors have read and agreed to the published version of the manuscript.

**Funding:** The initial motivation for this research originated from urban tree research within the DFG (German Science Council) funded research group FOR2423. We gratefully acknowledge the DFG project funding.

**Acknowledgments:** We are grateful to the Chinese Scholarship Council (CSC) for supporting the first author with a PhD scholarship.

**Conflicts of Interest:** The authors declare no conflicts of interest.

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
