# Peer review of "Towards Tree Green Crown Volume: A Methodological Approach Using Terrestrial Laser Scanning"

_remotesensing, doi:10.3390/rs12111841_

Round 1

Reviewer 1 Report

The authors are dealing with an interesting, undoubtedly interesting topic, but I am not sure if this is such a revolutionary idea that it deserves the publication of preliminary results. If it would be completed in full paper, I would have no problem with that at all.

Some notes and comments
TOF is a strange abbreviation - I wouldn't use it, it evokes Time of Flight sensors
Line 41 - In reference 7,  I did not find any mention of these artists in Basel. Check this reference
Describe crown voxelization and other techniques. Overall, Introduction seems to mean that the authors did not sufficiently examine the existing literature to be able to say we found a gap in knowledge.

Line 82, 83 - no need to list PC assembly, because of fast development. Nowadays, 3.6 GHz is no longer a high-performance processor clock rate. In addition, you do not specify the number of cores, graphics card, etc. In short: you can delete the sentence.

Describe k-means clustering in more detail. I understand the principle of this method, but in comparison with preprocessing description, where you list specific tools in Cloudcompare, there is a lack of libraries description used for k-means and other information needed to replicate the study. It is not even entirely clear that R was used. You mention the use of R on line 111, but not in the paragraph on lines 132-143.

Author Response

Dear reviewer,

We are very grateful for your encouraging and supportive comments. This is a first pilot study of tree green crown, and the feedback is extremely important to further develop our idea. We have revised our manuscript according to the comments and suggestions.

Please see the attachment, a point-to-point response.

Thank you again for your time and efforts!

Best regards

Reviewer 2 Report

Dear Authors

Initially, I congratulate you on the article presented. This may contribute to the proposed theme using TLS observations associated with field data. My suggestions are intended to improve the understanding of the work and facilitate its replication by other researchers. I advise the following revisions/improvements:

- Revise the introduction as it presents pieces of text that are confusing or that must be detailed.

- Some stages of the methodology should be improved;

- The discussion should be reviewed and improved and

- Evidence of the conclusions obtained by the study.

Comments are noted in the original text to assist them in implementing the suggestions.

Congratulations on the work.

Best regards!

Author Response

Dear reviewer,

We appreciate your thoughtful suggestions. The introduction has been revised accordingly. We also provide more detailed descriptions of our approach. As you advised, we have improved the discussion and conclusion.

Please see the attachment, a point-to-point response.

We would like to thank you again for your time and efforts.

Best regards

Reviewer 3 Report

The manuscript concentrates on defining and exemplifying the calculation of Tree Green Crown Volume (TGCVol), which is suggested as an improved 3D-based measure of tree foliage. Related measurement methodology is described, and results are shown for a set of scanned and analyzed solitary urban trees.

As such, TLS offers powerful assistance for evaluating the canopy layer of trees, and enables new measurement strategies. As I understand, the authors are proposing the described methodology to augment, or replace, traditional indices like LAI. LAI, however, has a solid definition of being principally the area of leaves per area of ground surface, which is highly rational given that the surface of leaves is responsible for the most essential processes such as photosynthesis. Compared to that, the proposed TGCVol is vaguely defined as crown volume “filled with foliage”, which is calculated as green clusters, derived from TLS points and found in the canopy layer.

Apart from definition, the TGCVol has certain shortcomings. First, the extraction methodology is not adequatly described. The abstract indicates that RGB values are an essential part in its extraction, but further in the manuscript it is indicated that RBG values were only used for manual fine tuning of the geometry-based classification. Second, as expected and also demonstrated, the outcome depends higly on the number of clusters, which prevents its easy standardization. Third, it's not only the number of clusters but also trees' structure, scanning resolution, number of scanning points and location, as well as wind which would make differences. And fourth, it's difficult to perceive this as a general methodological guideline for anything else than solitary trees, thus narrowing drastically its applications.

This criticism should not be understood as suggesting TGCVol to be useless; it may prove to be useful for various measurements. I must however conclude that the provided description on the TGCVol does not satisfy the definition of technical notes, being "significant advances and novel aspects of experimental and theoretical methods and techniques". Rather, I would see it as a case-sensitive part of another study enroute towards applied outcomes. To become a distinguished stand-alone methodological description, it must at least evolve from its present form.

One final remark: try to avoid excess use of quotation marks when they are not necessarily needed, as they tend to make reading a bit messy.

Some more detailed comments can be found below.

Abstract:
- rows 10-11: In the first sentence you refer to crown volume and its functions. Photosynthesis, matter production and shadowing could be seen as functions, but I'm not quite confident about storm resistance and being a habitat. I consider them as related, but not "functions" deriving from the crown volume. Maybe you can consider rephrasing the sentence.

Introduction:
- row 56: does "green surface" refer to tree canopy?

Materials and methods:
- Figure 1: first, try to take some background map which is readable at the normal page size, OpenStreetMap may not be the best option; second, don't use reference scale but rather scale bar, as most people won't be reading this as a default size print; third, use generally understandable coordinate system, not projected to something that is not defined; and fourth, I would recommend removing the word "Legend" from the legend, as it's rather self-evident
- row 70: rather than saying that 26 trees cover a wide range of species, would be more informative to tell the species (later noticed it comes in Table 2, but some information at this step would be useful)
- rows 72-74: I think it's kind of self-evident that these trees at roadsides and parks are not naturally developed, at least in the sense of being comparable to trees of same species and similar size growing in a forest
- Table 1: how is "lowest leaf layer" defined, does it refer to the first grean leaf found when advancing upwards from the ground, or somehow relates to the lowest living branch?
- row 94 onwards: is the scanner recording RGB values separately after the normal scan? In that case, wind conditions are also important to be mentioned, given that any movements in the foliage could affect the results.
- row 102: conventionally, scanning is defined either as single-scan or multi-scan; does multi-single-scan differ from these?
- row 104: I wouldn't think you can capture "complete crown surface" in any circumstances, and additionally, I think your primary interest is in the green parts inside the crown and not only the surface
- row 115: here, "filled with foliage" appears again. The definition however is vague (particularly as given in quotation marks), does it simply refer to the combined volume of leaves?
- row 122 onwards: so does this mean that CANUPO uses only geometry but not RGB information? How was the separation trained, as there is no description of the process?
- row 136: given that k-means clustering is relatively basic procedure and explained above, there's no need to explain that with k=1 all the elements belong to one cluster
- row 140: why was k=1400 selected?
- row 153: is reaching a value 1 even theoretically possible, given that even uniformly distributed leaves would have spaces between them, which would result in at least minor gaps between k>1 clusteres

Results:
- Table 2: I'm wondering if any of the readers is truly interested in specific statistics regarding to every studied tree; perhaps you could consider consolidating the main messages into a shorter table which would be easier to digest

Discussion:
- rows 182-183: to specify "work towards methodological elements of an assessment protocol and to specify further methodological issues" as one of the most important outcomes does not sound too convincing
- row 189: I think that the extraction of individual trees is not the primary challenge in most conditions, but the fact that nearby trees are hindering the scanning visibility to a level which sounds unbearable for the methodology suggested in the manuscript
- rows 196-197: given that solitary trees, scanned in spacious environment with six scans per tree, will require manual fine tuning to gain the desired result, I'm not too confident of the applicability of the proposed methods to more difficult conditions
- row 199: now you tell that segmentation failure was about RGB values, before you told that segmentation method characterizes features according to their geometry (row 123; i.e. excluding spectral information), so which actually is the case?
- row 200: which is the well-recognized problem you refer to; that automatic segmentation fails, or RGB exposures are not consistent? In addition, with all the respect to the cited reference [20], there would be avaible a number other options which would be much more suitable to the discussed theme
- rows 201-202: here RGB data is emphasized as a core issue, but no details of its use were provided before, which makes relatively difficult to follow the study
- rows 208-209: this would be the general outcome of convex hull type approaches when dealing with irregular shapes, and nothing too specifically related to this study
- row 212: "running the risk to compare apples and oranges" would not necessarily be the most precise way to describe these problems
- row 226: connections of these results to habitat suitability or heat mitigation remain somewhat unclear, this would require some clarification

Author Response

Dear reviewer,

We appreciate your extremely helpful and inspiring suggestions. These comments provided valuable insights for our methodological development. Our study is a first pilot study of tree green crown volume TGCVol; we understand we are facing the challenge of its definition and measurement (the same, by the way, for other crown variables such as wood volume, leaf area, crown volume, crown projection area, etc.). And on top of the measurement challenges comes the fact, that “true values” are virtually impossible to determine (at the standing tree and non-destructively).

We proposed a basic definition of TGCVol: The criterion of being a part of the green crown (green crown unit) is defined by the minimum number green hits (as a proxy for foliage) within a given irregular volume space (not voxel); and the green crown volume is the sum of the volume of all green crown units. Following this definition, the combined “k-means and convex hull” approach was used as a proxy to form green crown units and assess their volume. We added a new section for the definition which can be found in the revised manuscript, page 4, line 94-109.

In this pilot study, we wanted to analyze the characteristics and feasibility of our approach: our approach can be applied for different crown shapes. A relatively simple method of extraction was used since it is not our major focus. The RGB information was used during the CANUPO classification procedure. We have emphasized the use of the RGB value and give more details about the classification in the revised manuscript, as suggested. Also combining k-means clustering and the convex hull method for crown assessment appears to be new; we have not found any corresponding reference. We are certainly aware of the remaining challenges of our method as the reviewer mentions and is concerned about. This is why we offer this study as a pilot study, starting with the simpler overall conditions of solitary trees.

We are grateful to receive the reviewer’s inspiring comments, which are very helpful and encouraging for further developing our approach.

Please see the attachment, a point-to-point response.

Best regards

Reviewer 4 Report

Dear authors

The article, as I understand it, is a description of a certain concept of measuring the green part of the tree crowns and a presentation of realised research to verify this concept. In my opinion, the presented concept has been verified on the basis of the tests carried out and the correct conclusions have been drawn.
There are two main problems in this approach:
  - as you have noticed yourself, manual intervention in removing branches is necessary, it's tedious and time-consuming operation, and it's a significant obstacle in the wider application of such an approach.
- the publication lacks an attempt to assess the completeness of obtaining foliage data; How can the foliage / bough density affect the correctness of the results of the proposed method?

Good luck developing your idea

Author Response

Dear reviewer,

We sincerely thank you for the assessment. Your comments are very helpful for improving our approach. Our point-to-point response (in black) to the detailed comments (in orange) is given below. 

Comment 1. as you have noticed yourself, manual intervention in removing branches is necessary, it's tedious and time-consuming operation, and it's a significant obstacle in the wider application of such an approach.

Response: Our study is a pilot study of tree green crown volume which we wanted to analyze the characteristics and feasibility of our approach. We agree that manually editing the points is time-consuming and the classification procedure can be improved in further study. As mentioned in the discussion, we are aware that there are many alternatives for woody-leaf points classification which help to improve our approach for further application.

Comment 2. the publication lacks an attempt to assess the completeness of obtaining foliage data; How can the foliage / bough density affect the correctness of the results of the proposed method?

Response: Important point! Thank you. However, as for many crown variables, true values can hardly be produced (non-destructively) For TLS, we received less information of the foliage from the upper part of the crown. However, we are not able to assess how complete our foliage data is, since there is no true/reference value to validate. Therefore, this is a challenge in all TLS studies where true data cannot be generated.

What can be sure of is that the denser foliage and branches will lead to stronger occlusion effect, and we will have less complete information from the tree crown (the internal and upper part). It could be very interesting and useful to study the impact of foliage and branches density for k-means clustering approach in our further work.

We wish to thank you again for your time and efforts!

Best regards

Round 2

Reviewer 1 Report

Comment 1. Ok, I agree. Still, I think it can be confusing in the context of Remote Sensing journal

Comment 2. Done

Comment 3. I am satisfied with the revised introduction

Comment 4. This was exactly what I wanted

Now the article looks much better. I recommend for publication.

Reviewer 2 Report

Dear Authors

   Thanks for sending the updated version with implemented suggestions. It is noticed that the most complicated parts of the previous version have been corrected and the work is better presented. Improvements in methodologies and results have enriched the document.

    I liked that you separated the discussions from the conclusions.

    Only on line 59: variables like crown density viz.or crown. What's viz.or?

    Congratulations for your paper!

Best Regards.

Reviewer 3 Report

Although the applicability of the proposed methods remains limited and there are still methodological issues to be solved or at least clarified, this manuscript can be seen as a pilot study for something which has later potential. Given the improvements made after the first revision, I recommend the manuscript to be accepted.